# IMPACT study on intervening with a manualised package to achieve treatment adherence in people with tuberculosis: protocol paper for a mixed-methods study, including a pilot randomised controlled trial

Helen R Stagg ![ORCID] ,[1] Ibrahim Abubakar,[2] Colin NJ Campbell,[3] Andrew Copas,[4] Marcia Darvell,[3] Robert Horne,[5] Karina Kielmann,[6] Heinke Kunst,[7] Mike Mandelbaum,[8] Elisha Pickett,[9] Alistair Story,[10] Nicole Vidal,[6] Fatima B Wurie ![ORCID] ,[2] Marc Lipman ![ORCID] [3,9]

For numbered affiliations see end of article.

**Correspondence to**
Dr Marc Lipman;
marclipman@nhs.net

## ABSTRACT

**Introduction** Compared with the rest of the UK and Western Europe, England has high rates of the infectious disease tuberculosis (TB). TB is curable, although treatment is for at least 6 months and longer when disease is drug resistant. If patients miss too many doses (non-adherence), they may transmit infection for longer and the infecting bacteria may develop resistance to the standard drugs used for treatment. Non-adherence may therefore risk both their health and that of others. Within England, certain population groups are thought to be at higher risk of non-adherence, but the factors contributing to this have been insufficiently determined, as have the best interventions to promote adherence. The objective of this study was to develop a manualised package of interventions for use as part of routine care within National Health Services to address the social and cultural factors that lead to poor adherence to treatment for TB disease.

**Methods and analysis** This study uses a mixed-methods approach, with six study components. These are (1) scoping reviews of the literature; (2) qualitative research with patients, carers and healthcare professionals; (3) development of the intervention; (4) a pilot randomised controlled trial of the manualised intervention; (5) a process evaluation to examine clinical utility; and (6) a cost analysis.

**Ethics and dissemination** This study received ethics approval on 24 December 2018 from Camberwell St. Giles Ethics Committee, UK (REC reference 18/LO/1818). Findings will be published and disseminated through peer-reviewed publications and conference presentations, published in an end of study report to our funder (the National Institute for Health Research, UK) and presented to key stakeholders.

**Trial registration number** ISRCTN95243114

**Secondary identifying numbers** University College London/University College London Hospitals Joint Research Office 17/0726.

National Institute for Health Research, UK 16/88/06.

## Strengths and limitations of this study

► Patient-centred, mixed-methods approach, based on a robust understanding of the evidence on social, cultural and personal factors that influence adherence to medication to treat tuberculosis (TB).

► Evidence and experience of adherence captured from a variety of perspectives from across the UK, including patients, their carers and healthcare workers.

► Generalisable patient population of individuals at risk of non-adherence across low TB incidence settings.

► Development of a pragmatic and easy-to-use tool that captures the best evidence on adherence and allows its application in the clinic setting.

► The study culminates in a pilot trial of the manualised intervention; a larger subsequent definitive trial is needed to test whether the intervention is efficacious and cost-effective beyond any initial conclusions regarding validity and feasibility derived from the pilot.

## INTRODUCTION

Against a background of rising tuberculosis (TB) in the 1990s and 2000s, the need for a comprehensive approach to TB control in England was deemed necessary by Public Health England and the National Health Service (NHS) England. In January 2015, these bodies jointly launched the 'Collaborative Tuberculosis Strategy for England 2015–2020'.[1] This seeks to reduce TB incidence, decrease health inequalities and ultimately contribute to international efforts to eliminate TB as a public health problem. Ensuring

that people can take all of their medication as prescribed is one of the strategy's priorities, as poor adherence to treatment for TB is a driver of worse patient outcomes,[2–9] increases the risk of transmission (due to delayed sputum culture conversion)[10] and can promote the development of drug resistance.[3 11–16] Subsequent National Institute for Health and Care Excellence (NICE) guidance noted the lack of robust TB research in this area.[17]

Barriers to optimal adherence to treatment for TB may occur for a number of reasons. These include

► Patient-related factors, including perceptions and beliefs.
► Cultural influences and current mental state.
► Structural economic factors and social support networks.
► Health service factors that include treatment complexity as well as accessibility of those services and the relationships patients develop with service providers.[18 19]

Non-adherence is not a single issue and may take various forms, for example, suboptimal implementation (skipping doses) or stopping treatment early (eg, as soon as a patient feels better.).[20]

Although a series of studies have been undertaken to define the population groups most at risk of non-adherence,[19] it is currently difficult, prior to starting medication, to identify who may struggle with taking treatment as prescribed. To date, methods to support treatment address some, but not all, of the important underlying reasons for poor adherence. For example, the WHO's recent focus on digital technologies reflects our attention on individual-level determinants of adherence and reminder/observation-based systems[21]; far less research has addressed the social and structural barriers to staying on TB treatment.

In the UK, the development of an intervention to support adherence to treatment that is sensitive to the individual's cultural background and social circumstances, and can be routinely delivered within the NHS, is critical. To this purpose, a mixed-methods, patient-centred, approach to the study of the modifiable factors that influence patients' adherence to treatment of TB is required.

## METHODS AND ANALYSIS
### Research question
Can a manualised package of intervention be developed to help overcome the social and cultural factors that lead to poor adherence to treatment in NHS patients in the UK with active TB?

### Aim
To develop, pilot and evaluate process and interim outcomes for an effective manualised intervention that improves the likelihood of adherence to treatment among NHS patients at risk of poor adherence due to social, cultural and structural factors.

### Objectives
1. Synthesise current knowledge on (1) determinants of adherence to treatment for TB and (b) interventions that can support adherence, with particular emphasis on social and cultural barriers (scoping review and conceptual framework).
2. Apply a conceptual framework of adherence endorsed by NICE guidelines (the perceptions and practicalities (PAPA) approach)[18] to elucidate and address the personal, sociocultural and health systems context, mechanisms and pathways of poor adherence among NHS patients with TB (formative research).
3. Develop a manualised intervention (a series of systematic actions applied on the basis of a patient's needs assessment) with multiple components that can identify (1) NHS patients most at risk of non-adherence, (2) the salient modifiable barriers and (3) the tailored support mechanisms required to meet individual patient needs by matching appropriate interventions to specific barriers, as recommended by NICE (Development of Intervention). A manualised intervention was considered to be a suitable approach to managing adherence in TB, as it will enable a set of measures to be applied consistently within different NHS settings that will aid both clinicians and patients throughout the treatment journey. The content of the intervention will use existing support measures, employed in a systematic and structured way, and may also include any new interventions that are developed in response to the formative research. It will be compared with normal care.
4. Pilot the intervention package in people at risk of poor adherence to define how the components work in combination and separately (pilot study).
5. Evaluate the process of implementation of this intervention through describing the challenges and facilitators in delivering the package as intended (fidelity and reach) and assessing the impact of the intervention through evaluation of adherence indicators (process evaluation).
6. Use the findings of the pilot to assess the costs of delivering the manualised intervention in an NHS setting and to guide development of a proposal for a full randomised controlled trial (RCT).

## STUDY DESIGN
This study uses a mixed-methods approach. There are six subsections, reflecting the six objectives: (1) scoping review and conceptual framework, (2) formative research, (3) development of intervention, (4) pilot study, (5) process evaluation and (6) cost analysis and future work. Although the different elements are described separately further, the research activities for each will overlap, and some will run concurrently. The full programme of work and relationships between subsections is shown in figure 1. This paper reflects protocol V.4.0 (26 September 2019).[22]

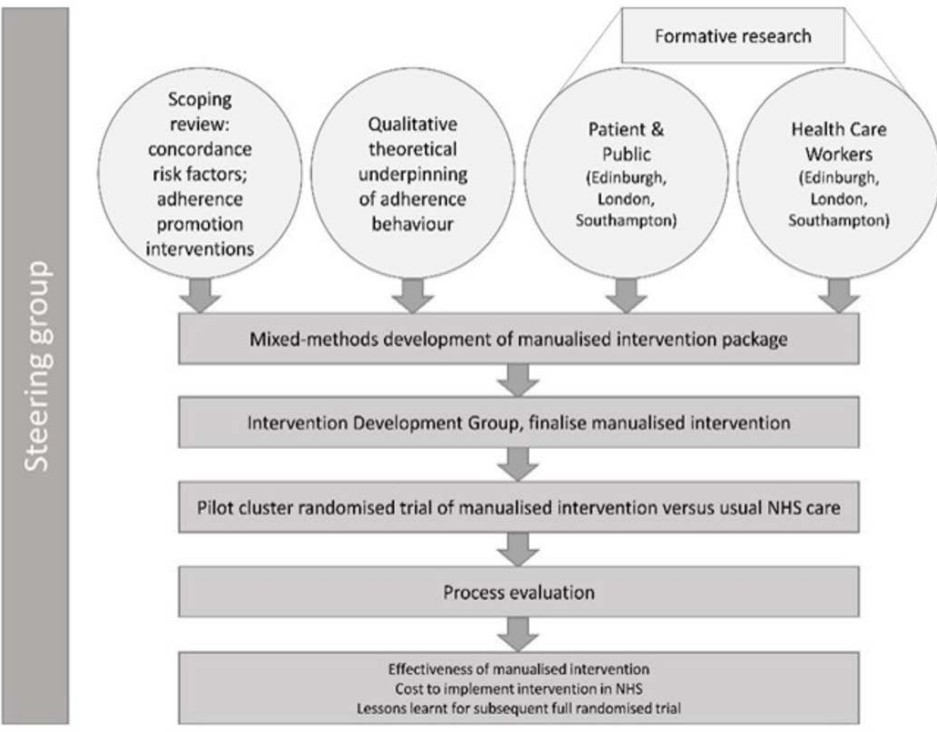

**Figure 1** IMPACT study. The six mixed-methods elements of the IMPACT study and the relationships between them. The study is guided by a steering group which has oversight over the entire process and culminates in a pilot study with associated evaluatory elements. IMPACT, Intervening with a Manualised Package to Achieve Treatment Adherence in People with Tuberculosis; NHS, National Health Service.

## Scoping review

To underpin the development of the manualised intervention, the first study subsection will undertake literature reviews to answer the following research questions:

1. What personal, social, cultural, health systems-related and structural factors affect individuals' ability to adhere to treatment for TB?
2. What kinds of intervention have been developed to address the multiple levels (personal, social, cultural, systems and structural) at which barriers to adherence may operate?
3. What is the evidence for the successful impact of interventions to address barriers to adherence to treatment for TB?

These questions will be answered through the three following reviews:

1. A scoping review of qualitative studies that examine the personal, social, cultural, health systems-related and structural factors affecting adherence to treatment for TB from the perspectives of adult patients, caregivers or healthcare providers, as well as studies that evaluate interventions to support adherence to treatment for TB. Data from all settings will be considered, but with a particular focus on low incidence, high income settings.
2. A critical review of quantitative studies examining the personal, social, cultural, health systems-related and structural factors affecting adherence to treatment for TB. This review focuses on low incidence, high in-

come, settings and observational study designs. Given the scoping nature of the review, findings will be descriptively analysed and not be stratified or disaggregated, for example, by site of disease. All relative and absolute measures of effect will be extracted.

3. A critical review of quantitative studies that have examined the effectiveness of interventions to improve adherence in people taking treatment for TB, building on existing systematic reviews, including the provision and delivery of information and/or education; enablers and/or incentives; social support; and case management approaches. Given the more limited literature, both observational studies and clinical trials will be included, from all settings. Findings will be descriptively analysed.

Together, these three reviews will enable us to conduct a critical interpretive synthesis of findings from qualitative and quantitative studies examining the assumptions and mechanisms of effect underlying interventions to improve adherence to treatment for TB. Further details on the reviews are available in the full protocol.[22]

## Formative research

### Formative research methods, recruitment and eligibility

In line with findings from the scoping review, we will develop topic guides for qualitative interviews exploring current models of TB service delivery in the study sites, providers' views on the barriers and facilitators for treatment adherence, and TB patients' and their family

members' experiences of starting and staying on treatment. Details about these guides are available within the full protocol.[22] The thematic analysis of these interviews will complement insights gained from the scoping reviews, which will allow us to refine a conceptual framework of adherence building on the PAPA.[18] Approximately 50 participants will be enrolled.

Adults (aged 18 and over) who are currently taking or recently completed treatment will be identified by the local TB service at four UK sites (Southampton, Edinburgh, Central London and East London)) and will be asked to take part in the study. The services have been chosen for their patient diversity, geographical spread and as reflecting the national TB picture. The patient group will be enriched with people who have been poorly adherent to treatment, although we will also include patients who report full adherence so that we can capture what may have enabled them to take treatment as prescribed. With patient consent, family members and/or carers will also be approached and asked to participate.

Health and social care workers from both primary and secondary care settings will be directly approached by researchers and invited to be interviewed. All those approached will be aged 18 or over and involved in TB management and care.

### Data collection methods

Three different methods of data collection will be used to undertake the formative research.

#### In-depth interviews: patients and family members/carers

These will be conducted as individual interviews with approximately 30 participants, using a topic guide. The areas to be explored are self-perception, personal beliefs and practices related to medicine-taking, health literacy and health-seeking behaviour, social support, cultural norms around health-seeking behaviour, and financial and other structural barriers.

#### Testing of Beliefs about Medicines Questionnaires (BMQ) and Brief Illness Perception Questionnaire (BIPQ) questionnaires with patients for their suitability

To ensure that the validated questionnaires (ie, [23]BMQ and the [24]BIPQ) are accessible and acceptable to patients, we will also conduct cognitive interviews with 10 patients. We will then have confidence to use them in the later pilot study to assess patient PAPA affecting adherence to anti-TB therapy. These will explore self-perception, personal beliefs and practices related to medicine-taking.

#### Semistructured interviews: healthcare providers

These will be undertaken with healthcare providers responsible for multiple aspects of TB care (doctors, nurses, social workers, directly observed therapy (DOT) providers, managers and administrators). They will focus on providers' perceptions of factors affecting patient understanding of TB and its treatment, service delivery models including staffing, organisation of care and

communication. We will interview four to six providers at each site, aiming for a total of 20 interviews.

We will use a framework approach[25] to facilitate initial analysis of the interview transcripts. Short patient case studies will be created for each patient interview. Data on health systems issues gained through mapping patient pathways and provider interviews will be organised using a deductive approach, with appropriate visual pathways.

### Development of the manualised intervention
#### Intervention development process

The data collected from the scoping reviews and formative research will be presented to, and synthesised by, an intervention development group (IDG). A manualised intervention will be developed to identify (1) NHS patients most at risk of non-adherence, (2) their salient modifiable barriers and (3) the tailored support mechanisms that meet individual patient needs by matching appropriate interventions to these specific barriers.

The IDG will aim to include
► Patients (in particular the homeless, ethnic minorities, migrants new to the UK, and patients with drug resistant TB).
► Family members/significant others of affected persons.
► Members of the public.
► Healthcare professionals (from both primary and secondary care).
► Other professionals who work with patients/communities affected by TB.

The constitution of the IDG will enable a coproduction approach to ensure that the intervention is pragmatic, can be delivered within the context of existing care pathways, and is of benefit to service users and those who are likely to access the intervention.

#### Intervention contents

The manualised intervention is likely to consist of a screening and assessment tool, in addition to a package of measures that can be tailored to the needs of individual patients from different population groups.

As patterns of adherence may be irregular over time, it is intended that the manualised intervention will be administered to all patients at each patient review. The tool thus needs to be quick and easy to administer. It may be electronically linked to patient records, enabling a comprehensive picture of the risk of possible non-adherence to be developed for each patient, as well as within a clinic population. The various delivery options for the intervention (such as using paper or an app) will be considered during its development stage.

The menu of supportive measures may, for example, include
► Informational intervention: for example, providing a convincing story setting out the rationale and ongoing need for medication, addressing concerns about potential adverse effects and consequences of treatment and what to do if such events occur (eg,

the participants will be informed that it is possible to change their treatment regimens to alternatives).[26]

► Practicalities and capability-based interventions: video-observed therapy (VOT); DOT; reminders, including text messaging; automated methods for monitoring and feedback, including electronic dosette boxes; use of a medication app; incentives, for example, financial and food vouchers; mitigation; and management of drug toxicity due to treatments.

► Social and system interventions: offering flexibility in appointments; enhanced guidance on 'navigating' clinic pathways; signposting patients to relevant services, for example, housing, drug and alcohol services, and social care; and providing peer support.

## Piloting the intervention
### Study design, recruitment and eligibility
Once the intervention is developed, proof of concept is required within the real world. This will be undertaken using a non-blinded cluster randomised pilot study that compares the manualised intervention to the usual standard of care in four London clinics treating TB. Two clinics will be randomly allocated to the intervention and two to standard of care. In the latter, the amount of support provided to patients is based on perceived need, as identified by a nurse-led review and a needs assessment. Most patients will have supported self-administered therapy, while others will be offered DOT and/or VOT if this is felt to be appropriate. We anticipate enrolment to commence in January 2020.

All consecutive patients aged 18 or over who are about to start treatment for TB, irrespective of the site of disease, will be approached to take part in the study. We will exclude individuals who are unable to provide informed consent, those already on treatment and those who are not expected to live for the duration of the study (a minimum of 6 months from starting treatment). Within the pilot study, it is essential to capture the entire treatment period for each patient in order to assess the effectiveness of the intervention. Due to the nature of the TB patient population in the UK, patients are likely to include people at greater risk of poor adherence, such as migrants newly arrived in the UK, people whose first language is not English, people with a mental health disorder, people taking immunosuppressive therapy or known to have immunodeficiency, those with a previous history of treatment for TB, or poor adherence with anti-TB medication, and people with a current or previous history of drug or alcohol misuse.

The purpose of the pilot study was not formal hypothesis testing. Given this, a target sample size of 80 patients enrolled (20 per site) was identified for the pilot study as providing useful information that can help determine whether the intervention will be deliverable within a clinical setting. It will also guide the development of a possible larger definitive study using the intervention. The four TB clinics of interest (in East and North London) each treat in excess of 60 relevant patients per annum.

Based on usage of DOT within the clinic populations seen at the treatment sites (ie, individuals currently identified as needing adherence support), we estimate that around 33% of patients will be at risk of non-adherence. Taking this as a minimum (as the manualised intervention is likely to be more sensitive than current risk assessments), we would expect that at least 26 of the 80 patients recruited will be identified as requiring adherence support. This sample size allows us to measure consent to enrolment for 80 individuals, data completeness for adherence and treatment outcomes for 80 individuals, data on acceptability and feasibility of the intervention package for around 40 individuals, and data on acceptability and feasibility of adherence support for at least 26 individuals (13 receiving the manualised intervention and 13 standard care).

### Outcomes of the pilot
The primary outcome of the study will be level of adherence, measured as the proportion of prescribed doses taken and assessed at 6 months from the start of treatment. In addition to this primary outcome, a number of secondary outcomes will also be measured, as follows:

1. Proportion consenting to the study.
2. Completeness of data for measures of adherence.
3. Proportion of patients withdrawing during the study and the reasons why.
4. Proportion of patients identified as needing adherence support in the intervention arm.
5. Proportion of patients offered adherence support and accepting it in the intervention arm.
6. Documentation of which adherence-promoting activities have been implemented among patients both in the standard of care and intervention arm, and when.
7. Detailed treatment implementation information: for example, proportion of patients completing treatment, proportion of patients still on treatment after 9 months or at study completion (whichever is the earlier).
8. Patterns of adherence (implementation and discontinuation).
9. Impact of manualised intervention on maintaining adherence over the duration of treatment.
10. Process variables: adherence-related PAPA.

### Measures of adherence
Our primary measure of adherence will be data obtained from medication monitoring boxes.[27] The boxes will not be set up act as a reminder system. Other measures will also be used and compared with this. These will include pill counts (the remaining medication in the box at the end of each month) and also patient-reported adherence, where we will ask patients to estimate how many doses they have missed in the last month. In the case of DOT or VOT methods being used, a record of missed doses will be kept.

### Administration of the manualised intervention

The patient's case manager (usually the TB clinic nurse), plus a study research nurse, will apply the intervention in partnership with the patient to identify whether personal, sociocultural and/or systems risk factors are present that suggest likely poor adherence with treatment. If these are identified, then the relevant measures outlined in the manualised intervention that may mitigate these will be reviewed and implemented with the agreement of the participant. These will be continued throughout the course of treatment or stopped if no longer deemed to be relevant or required on reassessment.

### Study schedule of visits

Study subjects will be seen at weeks 0, 2, 4, 8, 12, 16, 20 and 24 (earliest date of treatment completion). Should they require ongoing treatment after 6 months, they will be seen as clinically indicated. At each review, adherence assessments will be performed, in addition to an assessment of PAPA, the completion of the BMQ and the EQ-5D-5L Quality of Life questionnaires[23 28] and the GAD-7[29] and PHQ-9[30] to assess anxiety and depression. The manualised intervention will be applied if the patient is attending a clinic that has been randomised to the intervention arm. All data will be collected by the research nurse using standardised forms.

### Follow-up

Most patients who do not have clinically important drug-resistant disease receive 6 months of treatment. To allow for treatment interruptions, patients within the pilot study will be followed up to either treatment completion or for a total of 9 months from starting anti-TB therapy.

### Analysis and interpretation

Where possible, univariable analyses will be undertaken to compare each outcome measure listed between study arms (intervention and control). For the primary outcome (adherence), the mean percentage value will be reported by arm (intervention and control) and histograms will be used to describe the distribution of values by study arm. Binary secondary outcomes will be reported by the proportion of individuals achieving the outcome within each arm. The need to adjust for clustering by site and clinical care provider will be assessed using the cluster summary method (a t-test) to compare the cluster means or proportions (as appropriate, between arms). An assessment of the balance in baseline characteristics between the study arms will also be conducted. If randomisation has failed to evenly distribute key characteristics (eg, age, sex, ethnicity or other factors identified as important during the scoping reviews), then the cluster means or proportions will be adjusted for these differences before applying the t-test. This two-stage approach to analysis is described by Hayes and Moulton.[31] We recognise that adherence data may be highly skewed and thus may require compensatory analytical approaches.

The analysis of the first three of our secondary outcomes will address the feasibility of a definitive trial following a similar design to the pilot. Analysis of secondary outcomes 4–6 addresses the intervention, and complements the process evaluation (see below). Analysis of the primary outcome and final secondary outcomes around treatment adherence and completion provides initial information—given the modest sample size—concerning the effectiveness of the intervention, and may assist the sample size calculation for the definitive trial. They can also offer an alert in the unlikely event that the intervention is harmful.

### Power calculation

Although we are undertaking a pilot study and thus the numbers enrolled are small, table 1 indicates the power of our primary analysis to detect a range of absolute increases in adherence (10%–30%) from a variety of baseline values (50%–90%).

### Process evaluation
### Evaluation method

We will evaluate the implementation process by analysing the challenges and facilitators in delivering the package. The impact of the intervention will be assessed by evaluation of adherence indicators. We will use the findings of the pilot to assess the costs of delivering the manualised intervention in an NHS setting and to guide development of a proposal for a full RCT.

The process evaluation will consist of a description of the process of intervention implementation. It will assess how well the manualised intervention achieves its intended aim compared with standard care.

We will consider
1. The fidelity of the intervention as delivered in comparison to how it was designed and envisaged.
2. The reach of the intervention (the proportion of the target group receiving it).
3. The barriers to facilitating implementation of the intervention and how these can be addressed.
4. The pre-existing factors that facilitated implementation.

**Table 1** Power calculation for the pilot study, given a sample size of 80 individuals (40 per arm), across a range of baseline levels of adherence and absolute increases in that level

| Baseline adherence | Absolute increase | Power |
| --- | --- | --- |
| 70 | 30 | 0.98 |
| 60 | 30 | 0.88 |
| 50 | 30 | 0.82 |
| 80 | 20 | 0.86 |
| 70 | 20 | 0.61 |
| 60 | 20 | 0.49 |
| 90 | 10 | 0.54 |
| 80 | 10 | 0.24 |
| 70 | 10 | 0.18 |

### Process measures, recruitment and eligibility

Process measures for each element of the package will be developed once the manual development has been completed and will be used to assess success. They will include acceptability, uptake and change in practice. We will work with patients and staff separately at all four London sites. We will interview 20 patients (5 at each study site, ie, 10 from each arm) and, if possible, 20 healthcare workers (5 at each site). The patients will be selected within each site using purposeful sampling of clinic lists of every patient with active TB, to enable us to reflect the demographic spread of patients.

Key outputs will include a qualitative evaluation of delivery, the development of a narrative description of the process of intervention implementation and maintenance, and a quantitative assessment of adherence-related PAPA within intervention and control groups.

We will invite participation in the process evaluation from patients enrolled in the pilot study or staff members treating patients at one of the four London sites also involved in delivering the pilot study. They will be included if they are aged 18 years or over and able to provide informed consent. This will include probing anticipated versus real-life delivery of the intervention.

### Cost analysis and future work

In order to generate realistic estimates of the cost of the intervention, cost data from the NHS perspective will be collected during the pilot study using a cost data collection tool used by health economists (the Client Service Receipt Inventory) modified for TB.[32]

After the pilot study and process evaluation have been completed, a final intervention package will be designed for use in a definitive RCT of the manualised package of interventions. The design of this final package will be based on the results of the process evaluation and the experience gained during the piloting of the intervention, modifying the definitive trial design and/or data collection accordingly.

### PATIENT AND PUBLIC ENGAGEMENT

As documented previously, patient representatives will sit on the IDG for the study. In addition, TB Alert, the UK's only national TB charity, has membership of the IDG. The role of the IDG, which will meet regularly throughout the study, is described in the section Development of the manualised intervention. At the end of the study, the IDG will be involved in commenting on the findings and contributing to the dissemination plan.

### ETHICS, SPONSORSHIP, CONTACT DETAILS AND DISSEMINATION

The study is sponsored by the Joint Research Office of University College London and University College London Hospitals. This study received ethics approval on 24 December 2018 from the Camberwell St Giles Ethics Committee (REC reference 18/LO/1818; Level 3, Block B, Whitefriars, Lewins Mead, Bristol, BS1 2NT, UK;+44 (0)207104 8204; NRESCommittee.London-CamberwellStGiles@nhs.net). Findings will be published and disseminated through peer-reviewed publications and conference presentations, published in an end-of-study report to our funder (the National Institute for Health Research, UK) and presented to key stakeholders.

For public enquiries, please contact Marcia Darvell, IMPACT Study Project Co-ordinator, Respiratory Medicine, The Royal Free London NHS Foundation Trust, UCL Medical School Building, Rowland Hill Street, London, NW3 2PF, UK;+4420 8016 8375; m.darvell@ucl.ac.uk

For scientific enquiries, please contact the Chief Investigator Professor Marc Lipman (The Royal Free London NHS Foundation Trust and University College London), Respiratory Medicine, The Grove Centre, The Royal Free London NHS Foundation Trust, Rowland Hill Street, London, NW3 2PF, UK;+4420 7472 6452; marclipman@nhs.net.

### DATA SHARING STATEMENT

The datasets generated during and/or analysed during the current study will be available on request in a deidentified format and after publication of study outcomes and associated permission from the funder. Requests for data should be directed to Professor Marc Lipman as per the contact details previously mentioned.

### CONCLUSION

Our study will develop and pilot a manualised intervention to improve adherence to treatment for TB in the UK using a mixed-methods, patient-centred and provider-informed approach. This will enable us to begin to understand what motivates patients' treatment behaviour, while ensuring deliverability within the NHS. Our work is based on a robust understanding of the evidence on social, cultural and personal factors that influence adherence, and the interventions that are most effective in addressing these. The study reflects the geographical spread of TB in the UK and captures not only patient and expert clinical and academic experience but also that of family and carers to develop the intervention. A key feature of the study is the coproduction of a pragmatic and easy-to-use tool that uses the best evidence on adherence and allows its application in the clinic setting in a dynamic and iterative way.

Although the final pilot study may be limited to a relatively small sample size, it is hoped that its broad patient-centred perspective will make a useful contribution to our understanding of, and ability to deal effectively with, the risks of non-adherence to TB treatment in a population that can find this challenging. As many of the factors influencing adherence are likely to be generalisable to patients with other conditions in both

high and low resource settings, this study also has the potential to inform adherence interventions in other disease areas.

## Author affiliations

[1]Centre for Global Health, Usher Institute, University of Edinburgh, Edinburgh, UK
[2]Institute for Global Health, University College London, London, UK
[3]UCL Respiratory, University College London, London, UK
[4]Centre for Pragmatic Global Health Trials, Institute of Global Health, University College London, London, UK
[5]UCL School of Pharmacy, University College London, London, UK
[6]Institute for Global Health and Development, Queen Margaret University Edinburgh, Edinburgh, UK
[7]Department of Respiratory Medicine, Queen Mary University of London, London, UK
[8]TB Alert, Brighton, UK
[9]Department of Respiratory Medicine, Royal Free London NHS Foundation Trust, London, UK
[10]Find&Treat, University College Hospitals NHS Foundation Trust, London, UK

**Contributors** ML, IA, RH, KK and HRS conceived of the work. HRS, IA, AC, RH, KK, MM, AS, NV, FW, HK, EP and ML designed the work. HRS, CNJC, MD, RH and KK drafted the manuscript. All authors critically revised the manuscript and gave final approval of the version of the protocol manuscript to be published. All authors agreed to be accountable for all aspects of the work in ensuring that questions related to the accuracy or integrity of any part are appropriately investigated and resolved.

**Funding** This work was supported by the National Institute for Health Research (NIHR) Health Technology Assessment Programme, UK grant number 16/88/06.

**Disclaimer** The views expressed are those of the authors and not necessarily those of the National Health Service, UK, the National Institute for Health Research or the Department of Health and Social Care.

**Competing interests** IA, MD, HK, KK, ML, MM, EP, AS and FW have no competing interests to declare. RH is supported by the National Institute for Health Research (NIHR, Collaboration for Leadership in Applied Health Research and Care, North Thames at Bart's Health NHS Trust and Asthma UK (AUKCAR). Speaker engagements with honoraria with the following companies: Abbvie, Amgen, Astellas, AstraZeneca, Biogen, Erasmus, Idec, Gilead Sciences, GlaxoSmithKline, Janssen, Merck Sharp Dohme, Novartis, Pfizer, Roche, Shire Pharmaceuticals and TEVA. RH is founding director of a UCL-Business spin-out company (Spoonful of Sugar Ltd) providing consultancy on treatment engagement and patient support programmes to healthcare policy makers, providers and industry. HRS reports grants from Medical Research Council, UK, and grants from NIHR, UK, during the conduct of the study; others from Korean CDC and Johnson and Johnson (makers of Bedaquiline), others from Latvian Society Against Tuberculosis (funding through Otsuka and Johnson and Johnson), outside the submitted work. CNJC reports personal fees from Public Health England outside the submitted work.

**Patient consent for publication** Not required.

**Provenance and peer review** Not commissioned; externally peer reviewed.

**ORCID iDs**
Helen R Stagg http://orcid.org/0000-0003-4022-3447
Fatima B Wurie http://orcid.org/0000-0003-4802-2308
Marc Lipman http://orcid.org/0000-0001-7501-4448

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
