## [Reviewer comments · BMJ Open]

ARTICLE DETAILS

TITLE (PROVISIONAL)	The IMPACT Study: Intervening with a Manualised Package to ACHieve treatment adherence in people with Tuberculosis. A protocol paper for a mixed methods study, including a pilot randomised controlled trial
AUTHORS	Stagg, Helen R.; Abubakar, Ibrahim; Campbell, Colin; Copas, Andrew; Darvell, Marcia; Horne, Robert; Kielmann, Karina; Kunst, Heinke; Mandelbaum, Mike; Pickett, Elisha; Story, Alistair; Vidal, Nicole; Wurie, Fatima; Lipman, Marc

VERSION 1 – REVIEW'

REVIEWER	Stephanie Law Research fellow Department of Global Health & Social Medicine, Harvard Medical School, U.S.
REVIEW RETURNED	23-Jul-2019

GENERAL COMMENTS	This is a very exciting study and I look forward to seeing the results from it. It's such an important endeavor as experimental trials to improve adherence are sorely lacking in TB. I just have some comments that hopefully you would find helpful: 1) It would be good to define "manualized intervention" early on, and explain why that is a suitable type of intervention for this population & context. 2) You mentioned a "package of measures that can be tailored to the needs of individual patients from different " and also provided some examples of what they may include. Are these existing support measures or they will also have to be developed as part of this intervention? Of is the idea for this manualized intervention to only include support measures that are already being provided to patients, but it's meant to be a systematic way to identify patients who are more at risk and to provide support as early as possible? This should be clarified in the description. 3) Please briefly explain this exclusion and why it is thought this should be done, and what biases it could introduce: "Those who are not expected to live for the duration of the study (a minimum of six months from starting treatment)." 4) Also, is there a rationale for only enrolling new patients on to the study and not patients already on treatment? (i.e. you could potentially restrict it to patients who have been on treatment for less than 2 months, for example.) This could help increase your
--

	sample size during the pilot period. If there is a rationale, it should be described in the text. 5) Please clarify the sampling method for the pilot intervention - is it all, consecutive patients? 6) Your method of analysis should account for (at least) clustering at the site-level, (even with small number of clusters it is possible - see for example: Daniel McNeish & Laura M. Stapleton (2016) Modeling Clustered Data with Very Few Clusters, Multivariate Behavioral Research, 51:4, 495-518, DOI: 10.1080/00273171.2016.1167008.) 7) You mentioned logistic regression analyses, but your primary outcome seems to be a continuous variable. If you are dichotomizing your outcome (level of adherence), how do you plan on doing that (i.e. what cut-off will be used)? 8) Line 55-56: "The patients will be selected within each site using purposeful sampling of clinic lists of every fourth patient with active TB." This is unclear: are you doing purposeful sampling (i.e. picking patients purposefully to maximum their characteristics/experiences, for example) or are you doing systematic sampling (i.e every fourth patient)? 9) Can you give examples of the cost data you would be collecting? What would be considered cost-effective, or rather, is there a threshold to decide on whether it will be feasible for a larger trial? 10) Consider including some community feedback meeting as part of your dissemination plan (or maybe handout of findings available at the sites), you could get additional comments from other patients not in the IDG for planning a larger future study. 11) Do you have measures of 'success', that is, thresholds or cutoffs to help you decide whether the intervention is 'effective' and therefore worthy of a larger trial? (e.g. if adherence increases by 10% in the intervention group) 12) For the background, do you have any information on what the 'baseline' adherence rate is for your population? Would be good to provide that to have a sense of how big a problem is, and the potential statistical power you'll have as a result of that prevalence. 13) Also can you describe identified adherence barriers/risk factors for your context in the published literature? 14) You might consider doing focus group discussions with the health care providers to maximize on input & minimize time (given their busy schedules). 15) Variability in the support provided - how are you going to monitor that? For example, if there is some TB education program that is provided, how do you monitor fidelity/quality of delivery? This would be important for integrity of the study and would be an important factor to consider that could be affecting your findings. 16) Data collection methods for the pilot study aren't clear. How will you be collecting patient characteristics to use in your analyses
--	--

	(i.e. confounders, etc.)? Who will be administering the validated questionnaires to the patients?
--	---

REVIEWER	Ramnath Subbaraman Tufts University School of Medicine, USA
REVIEW RETURNED	16-Aug-2019

GENERAL COMMENTS	This is an important and timely research protocol. Many TB experts are revisiting the problem of TB medication adherence; however, most of the focus of recent research has been on use of digital adherence technologies, rather than on understanding the broader contextual factors that shape medication adherence. Few people have focused on studying these broader contextual factors or on developing interventions to address them. This research protocol is more of a research and intervention development process, with multiple stepwise research agendas embedded within the same protocol. I appreciate their mixed methods approach to intervention development and pilot testing, which is a rigorous way to go about developing a multi-component intervention. Overall, I think that this protocol will be very helpful to researchers in the TB community, in helping others think about systematic stepwise, informed approaches to intervention development in their own contexts. I think this is somewhere between a major and minor revision based on the feedback below - they just need to flesh out some more detail to make this more helpful to readers. My main critique of this submitted research protocol is that, while the general stepwise nature of the process is clear, many of the steps lack sufficient detail to fully understand the scope and limitations of the specific step. As such, I think many readers may not have sufficient detail to understand exactly how each research step will be executed. More detail on the following steps would be helpful:  1. Research question: In addition to the over-arching question posed regarding intervention development that has been articulated, it seems to me that there is a preliminary question the authors are trying to answer in Objectives 1 and 2, which is: What are the determinants of medication non-adherence among patients with TB in the NHS? (or something similar). 2. The Scoping Review: Many details are lacking to describe the scope of these reviews. For example, are these scoping reviews limited to low TB incidence settings (which might be more relevant to the authors' goals)? What kinds of quantitative studies are they hoping to include? (E.g., Cohort studies? Survey-based studies?) What kinds of data are they hoping to extract from these quantitative studies? (E.g., odds ratios or risk ratios from cohort studies? proportions from survey-based studies?). Will they stratify or disaggregate findings for different types of patients? (e.g., drug-susceptible / drug-resistant or pulmonary / extrapulmonary)? What approach are they using to extract qualitative findings and analyze them (e.g., meta-ethnography, etc.)? I understand that there is only so much detail that the authors will provide about each of their research steps, since this is a larger
---

step-wise research process. I also understand that these are "scoping" or "critical" (rather than systematic) reviews. However, since they are choosing to publish this protocol, it would at minimum be helpful for readers to understand the contours of their research goals and to get some basic sense of their methodology.

3. Formative Research:

Will the researchers stratify patients (or ensure inclusion) of patients across both intensive and continuation phases of TB therapy? And if so in what proportion in their initial sampling? This is important because anecdotal evidence suggests that patients have very different medication adherence in the intensive phase (when they're still feeling sick) and the continuation phase (when they have often had improvement in symptoms).

Is there any plan to collect more patient or healthcare provider interviews if thematic saturation is not achieved? It is common to anticipate that you might have more or less interviews in qualitative research depending on how many new themes are arising with progressive coding of interviews.

For the in-depth interview of patients and/or their families, will you also ask them about their perspectives on the health system as a factor affecting their medication adherence?

It would be nice to include a table or box with examples of the rough types of questions you will be asking patients / families for each qualitative domain (i.e., self-perception, personal beliefs, health literacy, etc.). Same for the healthcare provider interviews

4. Cognitive assessments: I'm not sure what cognitive assessments or interviews entail - could the authors describe this better or provide citations.

5. Analysis of the formative research: The authors do not provide any detail on their general research method for coding and analyzing interview transcripts. Will they use an inductive approach? Will they use a deductive approach based on a pre-existing framework?

6. Intervention contents: The authors do not mention addressing mental health (especially major depression or anxiety) or addressing stigma as part of their potential intervention contents. While I know the intervention ideas will emerge or gain support from the formative research, I think that stigma and mental health are important enough in the prior TB literature to merit a few sentences of discussion here.

7. Intervention piloting: A few sentences on the current standard of care at the possible treatment sites researchers will select from would be helpful. Do these clinics currently put everyone on in-person DOT? Or do many of them use self-administered therapy (i.e., patient takes pills on their own at home)? Or have any of these centers adopted video DOT?

While this is only a pilot study, there may be some basis for conducting a power calculation based on their primary outcome of the proportion of prescribed doses taken (i.e., dosing implementation). I think the key question is: what percentage difference between two groups do they think they might have

	power to detect with the current sample size? I understand that the goal of this study is NOT to answer a hypothesis, but the preliminary information from these pilot trials are often crucial for estimating sample size for much larger, more definitive clinical trials, so understanding what percentage of difference the current sample size could detect for a given power would be helpful. 8. Urine testing: The authors note that they will obtain a urine sample for testing using an assay based on the type of regimen the patient is taking. This begs the question: will all patients be drug-susceptible TB patients or will they include drug-resistant TB patients also? Also, can they describe what assay they would use at minimum for drug-susceptible TB patients? This is important - INH based urine tests detect any dose taken in the prior 48 to 72 hours, while rifampin based tests often only detect doses taken in the prior 12+ hours. So it is possible to have a negative rifampin-based urine test result even if the patient is adherent (took their pills the prior day). Also, a few sentences of discussion of the limitations of their urine measurement approach would be helpful.
--	---

VERSION 1 – AUTHOR RESPONSE

Reviewer: Stephanie Law

1) It would be good to define "manualized intervention" early on, and explain why that is a suitable type of intervention for this population & context.

We have added new text under Objective 2 to define a manualised intervention (line 143) and explain why this has been chosen as a suitable intervention (line 148).

2) You mentioned a "package of measures that can be tailored to the needs of individual patients from different" and also provided some examples of what they may include. Are these existing support measures or they will also have to be developed as part of this intervention? Or is the idea for this manualized intervention to only include support measures that are already being provided to patients, but it's meant to be a systematic way to identify patients who are more at risk and to provide support as early as possible? This should be clarified in the description.

Thank you, we have addressed this in the new text under Objective 2 (line 150).

3) Please briefly explain this exclusion and why it is thought this should be done, and what biases it could introduce: "Those who are not expected to live for the duration of the study (a minimum of six months from starting treatment)."

We have now addressed this under 'Piloting the Intervention' (line 327), where we have added a new sentence to explain that we wish to follow patients from start of their treatment onwards. This is critical in order to assess the impact of our intervention across the entire span of the treatment period, given that different factors will come into play at different stages of the patient experience. Whilst we agree that patients who may die during treatment could be those who are least adherent with therapy (i.e. a potential target population for the intervention), within the pilot study it is important to collect data for as long as possible on each subject. In addition, many patients who die during treatment will have multiple co-morbidities (e.g. cancer) that are likely to be the major determinant of outcome rather than the societal factors of non-adherence being explored in this study.

4) Also, is there a rationale for only enrolling new patients on to the study and not patients already on treatment? (i.e. you could potentially restrict it to patients who have been on treatment for less than 2 months, for example.) This could help increase your sample size during the pilot period. If there is a rationale, it should be described in the text.

Please see point 3) above.

5) Please clarify the sampling method for the pilot intervention - is it all, consecutive patients?
Thank you for noting this lack of clarity. All consecutive patients will be approached within the pilot study (line 324).

6) Your method of analysis should account for (at least) clustering at the site-level, (even with small number of clusters it is possible - see for example: Daniel McNeish & Laura M. Stapleton (2016) Modeling Clustered Data with Very Few Clusters, *Multivariate Behavioral Research*, 51:4, 495-518, DOI: 10.1080/00273171.2016.1167008.)

Thank you for this helpful point. We are considering clustering at both the site and healthcare provider level. This is now clarified in lines 427 and 433.

7) You mentioned logistic regression analyses, but your primary outcome seems to be a continuous variable. If you are dichotomizing your outcome (level of adherence), how do you plan on doing that (i.e. what cut-off will be used)?

We thank the reviewer for noticing this important point. They are correct, we are analysing adherence as a continuous, not a binary, outcome. The description of the analysis has been altered in the text (line 417) and we have removed references to logistic regression.

8) Line 55-56: "The patients will be selected within each site using purposeful sampling of clinic lists of every fourth patient with active TB." This is unclear: are you doing purposeful sampling (i.e. picking patients purposefully to maximum their characteristics/experiences, for example) or are you doing systematic sampling (i.e. every fourth patient)?

We apologise for the confusion, this will be purposeful sampling that enables us to reflect the demographic spread of patients with tuberculosis (lines 494).

9) Can you give examples of the cost data you would be collecting? What would be considered cost-effective, or rather, is there a threshold to decide on whether it will be feasible for a larger trial?
New wording has been added which mentions the Client Service Receipt Inventory (CSRI) through which we will collect cost and encounter data for the study patients lines 510.

10) Consider including some community feedback meeting as part of your dissemination plan (or maybe handout of findings available at the sites), you could get additional comments from other patients not in the IDG for planning a larger future study.

Thank you for this suggestion; it will be added to our dissemination plan for the study.

11) Do you have measures of 'success', that is, thresholds or cut-offs to help you decide whether the intervention is 'effective' and therefore worthy of a larger trial? (e.g. if adherence increases by 10% in the intervention group)

As indicated within our power calculation (line 451), we now consider 10-30% increases in adherence to be important enough to warrant a full trial.

12) For the background, do you have any information on what the 'baseline' adherence rate is for your population? Would be good to provide that to have a sense of how big a problem is, and the potential statistical power you'll have as a result of that prevalence.

Accepting that this is a pilot study, we have provided a table (Table 1, line 451) showing how study power varies at a range of different adherence levels for the intervention and control groups.

13) Also can you describe identified adherence barriers/risk factors for your context in the published literature?

Yes, this is included within the second scoping review (line 193).

14) You might consider doing focus group discussions with the health care providers to maximize on input & minimize time (given their busy schedules).

Thank you for this suggestion. Having spoken to our colleagues in various hospital, we do not anticipate a one-to-one approach to be an issue, but we will bear this in mind.

15) Variability in the support provided - how are you going to monitor that? For example, if there is some TB education program that is provided, how do you monitor fidelity/quality of delivery? This would be important for integrity of the study and would be an important factor to consider that could be affecting your findings.

This element of intervention delivery will be monitored during the Process Evaluation through (line 481; 504). Of note, within both usual care and intervention arms, there is a level of education on tuberculosis that is provided to all patients. We would expect this to be standardised within the control and intervention groups (all tuberculosis services use the Royal College of Nursing UK TB Case Management Tool).

16) Data collection methods for the pilot study aren't clear. How will you be collecting patient characteristics to use in your analyses (i.e. confounders, etc.)? Who will be administering the validated questionnaires to the patients?

We will develop and use a standardised data collection sheet (Case Report Form), which will be administered by a research nurse (line 409). Some data may be obtained from clinical notes.

Reviewer: Ramnath Subbaraman

1. Research question: In addition to the over-arching question posed regarding intervention development that has been articulated, it seems to me that there is a preliminary question the authors are trying to answer in Objectives 1 and 2, which is: What are the determinants of medication non-adherence among patients with TB in the NHS? (or something similar).

We agree with the reviewer and apologise for this not being clear in our original text. This element is captured within objective 1. We have rephrased the research question to more closely match the reviewer's suggestion (line 126)

2. The Scoping Review: Many details are lacking to describe the scope of these reviews. For example, are these scoping reviews limited to low TB incidence settings (which might be more relevant to the authors' goals)? What kinds of quantitative studies are they hoping to include? (E.g., Cohort studies? Survey-based studies?) What kinds of data are they hoping to extract from these quantitative studies? (E.g., odds ratios or risk ratios from cohort studies? proportions from survey-based studies?). Will they stratify or disaggregate findings for different types of patients? (e.g., drug-susceptible / drug-resistant or pulmonary / extrapulmonary)? What approach are they using to extract qualitative findings and analyze them (e.g., meta-ethnography, etc.)?

Thank you. The protocol paper is an abridged version of a much longer protocol approved by NIHR; further details on our scoping review methodology can be found at the link now provided at the end of the paper <https://www.fundingawards.nihr.ac.uk/award/16/88/06> We have now added further details into the text (line 190 onwards).

3. Formative Research: Will the researchers stratify patients (or ensure inclusion) of patients across both intensive and continuation phases of TB therapy? And if so, in what proportion in their initial sampling? This is important because anecdotal evidence suggests that patients have very different medication adherence in the intensive phase (when they're still feeling sick) and the continuation phase (when they have often had improvement in symptoms).

Thank you for this comment. While we recognise that patterns of adherence can vary over the course of treatment, it may be difficult to elicit the views of patients in the intensive phase of treatment given

their state of health and well-being. Pragmatically, we will ask each patient interviewed to reflect on changes in patterns of adherence over the course of their treatment journey.

Is there any plan to collect more patient or healthcare provider interviews if thematic saturation is not achieved? It is common to anticipate that you might have more or less interviews in qualitative research depending on how many new themes are arising with progressive coding of interviews. Our sample of patients will be selected to allow for maximum diversity in terms of the characteristics of patient populations at the four sites. Given the known differences in the models of care in the four study sites, the sex, age, and ethnic composition of the patient population, we do not anticipate thematic saturation. For health provider interviews, especially for those conducted within a site, saturation is likely, and the number of interviews will be tailored accordingly.

For the in-depth interview of patients and/or their families, will you also ask them about their perspectives on the health system as a factor affecting their medication adherence?

Yes, we will ask about this. The questions are available in the topic guide of the protocol on the NIHR Website <https://www.fundingawards.nihr.ac.uk/award/16/88/06>

It would be nice to include a table or box with examples of the rough types of questions you will be asking patients / families for each qualitative domain (i.e., self-perception, personal beliefs, health literacy, etc.). Same for the healthcare provider interviews

This is available in the full protocol published on the NIHR Website. We now link to this protocol within the manuscript (line 559).

4. Cognitive assessments: I'm not sure what cognitive assessments or interviews entail - could the authors describe this better or provide citations.

Thank you, we appreciate that this was confusing, and have now clarified this to make it clear that the cognitive testing is a 'sense check' of the BMQ and BIPQ (line 246).

5. Analysis of the formative research: The authors do not provide any detail on their general research method for coding and analyzing interview transcripts. Will they use an inductive approach? Will they use a deductive approach based on a pre-existing framework?

We will use a framework approach (Gale et al. 2013) to facilitate initial analysis of the interview transcripts. The framework is organised around the five domains of determinants of adherence behaviour as identified by existing literature (individual; social; health-systems related; treatment-related; structural). At the same time, short patient case studies will be created for each patient interview to maintain the integrity of the narrative and privilege the subjective, lived experience of having tuberculosis and being on treatment for the conditions. Data on health systems issues gained through mapping of patient pathways and provider interviews will be more structured and will be organized using a deductive approach and appropriate visual display, for example, through flow-charts and decision-making trees.

Text has been added to the paper to clarify this (line 261).

Ref: Gale NK, Heath G, Cameron E, Rashid S, Redwood S. Using the framework method for the analysis of qualitative data in multi-disciplinary health research. *BMC Med Res Methodol.* 2013;13:117 doi:10.1186/1471-2288-13-117

6. Intervention contents: The authors do not mention addressing mental health (especially major depression or anxiety) or addressing stigma as part of their potential intervention contents. While I know the intervention ideas will emerge or gain support from the formative research, I think that stigma and mental health are important enough in the prior TB literature to merit a few sentences of discussion here.

Thank you for this helpful suggestion, this was an oversight on our part. We have now added GAD-7 and PHQ-9 assessments to assess anxiety and depression (line 407). We will also address mental health interventions in our manual if detected using the intervention screening tool.

7. Intervention piloting: A few sentences on the current standard of care at the possible treatment sites researchers will select from would be helpful. Do these clinics currently put everyone on in-person DOT? Or do many of them use self-administered therapy (i.e., patient takes pills on their own at home)? Or have any of these centers adopted video DOT?

The study sites have been chosen to reflect the level of care found in two different tuberculosis Services. Care is provided in line with the Royal College of Nursing UK Tuberculosis Case Management Tool and is compliant with 2016 UK NICE guidance. The amount of support provided to patients is based on perceived need, as identified by tuberculosis nurse review and a needs assessment. Most patients will have supported self-administered therapy, whilst others will be offered DOT and/or VOT if this is felt to be appropriate. This is now described in the text on line 321.

While this is only a pilot study, there may be some basis for conducting a power calculation based on their primary outcome of the proportion of prescribed doses taken (i.e., dosing implementation). I think the key question is: what percentage difference between two groups do they think they might have power to detect with the current sample size? I understand that the goal of this study is NOT to answer a hypothesis, but the preliminary information from these pilot trials are often crucial for estimating sample size for much larger, more definitive clinical trials, so understanding what percentage of difference the current sample size could detect for a given power would be helpful. Thank you for this comment. As discussed above we have now included a table with varying differences between baseline (control arm) and intervention adherence (line 451).

8. Urine testing: The authors note that they will obtain a urine sample for testing using an assay based on the type of regimen the patient is taking. This begs the question: will all patients be drug-susceptible TB patients or will they include drug-resistant TB patients also? Also, can they describe what assay they would use at minimum for drug-susceptible TB patients? This is important - INH based urine tests detect any dose taken in the prior 48 to 72 hours, while rifampin-based tests often only detect doses taken in the prior 12+ hours. So it is possible to have a negative rifampin-based urine test result even if the patient is adherent (took their pills the prior day). Also, a few sentences of discussion of the limitations of their urine measurement approach would be helpful.

We agree with your reservations about urine testing, and have removed all reference to urine testing from our protocol. Instead, we now plan to use medication monitor boxes as a tool to measure adherence (but not a reminder system). These will become our primary outcome measure, with pill counts and self-reported adherence as secondary measures. We will triangulate between these measures. This has been altered in the text in line 379. The change also enables us to use the same measures of adherence irrespective of the drugs being taken by patients to treat their tuberculosis.

The changes we have made to the manuscript in line with the reviewers' comments have substantially altered some parts of the document. We believe that it is considerably improved by this. However, we will need to ensure that our funders NIHR are happy with these. Given the tight turnaround time required for the manuscript, we are still awaiting confirmation from NIHR at the time of resubmission of these revisions. We will keep you informed of NIHR's decision; and in the meantime hope that you find our responses to the reviewers, and changes in the paper, satisfactory.

VERSION 2 – REVIEW

REVIEWER	Stephanie Law Department of Global Health and Social Medicine, Harvard Medical School, Boston
REVIEW RETURNED	23-Sep-2019

GENERAL COMMENTS	Thank you for responding to all the comments and making changes accordingly; the changes were all very clear. Just one minor comment remains - what is meant by "this" on Line 418: " Binary secondary outcomes will be reported by the proportion of individuals achieving this within each arm." I look forward to seeing the future results from the study.
--

REVIEWER	Ramnath Subbaraman Department of Public Health and Community Medicine, Tufts University School of Medicine
REVIEW RETURNED	23-Sep-2019

GENERAL COMMENTS	I have reviewed the authors' revisions to this protocol, and they have done a careful job of addressing my concerns by providing more details on this study. Especially helpful is the link to their full protocol, which will be helpful to others who want to take a deep dive into understanding their methods. This protocol will be a very helpful contribution to others doing research on TB medication adherence.
---

VERSION 2 – AUTHOR RESPONSE

Reviewer: Stephanie Law

1. Thank you for responding to all the comments and making changes accordingly; the changes were all very clear.

Thank you for this positive comment.

2. Just one minor comment remains - what is meant by "this" on Line 418: " Binary secondary outcomes will be reported by the proportion of individuals achieving this within each arm."

We have now clarified this on line 408.

Reviewer: Ramnath Subbaraman

3. I have reviewed the authors' revisions to this protocol, and they have done a careful job of addressing my concerns by providing more details on this study. Especially helpful is the link to their full protocol, which will be helpful to others who want to take a deep dive into understanding their methods. This protocol will be a very helpful contribution to others doing research on TB medication adherence.

Thank you for this positive comment.

The changes we have made to the manuscript in line with the reviewers' comments have substantially altered some parts of the document. We believe that it is considerably improved by this. However, we will need to ensure that our funders NIHR are happy with these. Given the tight turnaround time required for the manuscript, we are still awaiting confirmation from NIHR at the time of resubmission

of these revisions. We will keep you informed of NIHR's decision; and in the meantime hope that you find our responses to the reviewers, and changes in the paper, satisfactory.